# Carbogen inhalation during non-convulsive status epilepticus: A quantitative exploratory analysis of EEG recordings

S. Ramaraju[1]*, S. Reichert[2], Y. Wang[1], R. Forsyth[3‡], P. N. Taylor[1‡]

**1** Interdisciplinary Computing and Complex Bio-Systems Group, School of Computing Science, Newcastle University, Newcastle upon Tyne, United Kingdom, **2** Department of Electrical Engineering, Medical Engineering and Computer Science, Offenburg University of Applied Sciences, Offenburg, Germany, **3** Institute of Neuroscience, Faculty of Medical Science, Newcastle University, Newcastle upon Tyne, United Kingdom

‡ These authors are joint senior authors on this work.
* sriharsha.ramaraju@newcastle.ac.uk

## Abstract

### Objective

To quantify the effect of inhaled 5% carbon-dioxide/95% oxygen on EEG recordings from patients in non-convulsive status epilepticus (NCSE).

### Methods

Five children of mixed aetiology in NCSE were given high flow of inhaled carbogen (5% carbon dioxide/95% oxygen) using a face mask for maximum 120s. EEG was recorded concurrently in all patients. The effects of inhaled carbogen on patient EEG recordings were investigated using band-power, functional connectivity and graph theory measures. Carbogen effect was quantified by measuring effect size (Cohen's d) between "before", "during" and "after" carbogen delivery states.

### Results

Carbogen's apparent effect on EEG band-power and network metrics across all patients for "before-during" and "before-after" inhalation comparisons was inconsistent across the five patients.

### Conclusion

The changes in different measures suggest a potentially non-homogeneous effect of carbogen on the patients' EEG. Different aetiology and duration of the inhalation may underlie these non-homogeneous effects. Tuning the carbogen parameters (such as ratio between $CO_2$ and $O_2$, duration of inhalation) on a personalised basis may improve seizure suppression in future.

**Data Availability Statement:** Data and code is available via Figshare: https://doi.org/10.6084/m9.figshare.13239719.v2.

**Funding:** This work was funded by Epilepsy Research UK to Dr R Forsyth(P1103) and Wellcome Trust (210109/Z/18/Z) to Dr P N Taylor.

**Competing interests:** NO authors have competing interests.

## 1. Introduction

Status epilepticus (SE) is a situation of continuing seizure activity or repetitive seizures without recovery lasting (by convention) over 30 minutes [1]. Morbidity and mortality are affected by factors including age, aetiology and time to first treatment [2, 3]. Non-Convulsive Status Epilepticus (NCSE) is a subtype of SE again of varied aetiology [4], and is characterised by more subtle SE without prominent motor signs but generally reduced awareness of surroundings and responsiveness [5]. NCSE is often observed in the context of pre-existing neurological conditions such as injury and neurogenetic syndromes associated with severe epilepsy or learning difficulties [6]. NCSE should be suspected in children with epilepsy who undergo an otherwise unexplained deterioration in behaviour, speech, memory, or school performance [6].

Many treatments for convulsive SE have anaesthetic or sedative properties. This is disadvantageous in NCSE as sedation can activate NCSE and children are at increased prior risk of respiratory problems [7]. The ideal treatment in this case would be immediate acting, non-drowsy, maintaining respiratory function and acting for a sustained period. Tolner et al. [8], showed that induction of temporary mild respiratory acidosis (reduction of pH levels) by inhaling carbogen (5% carbon dioxide and 95% oxygen) has anticonvulsant action, and this would be of potential value in NCSE. Carbogen has shown to terminate acute onset seizures in rats, non-human primates and adult humans [9–15].

During NSCE, electroencephalographic (EEG) activity is abnormal. Patient EEG dynamics can include pathological polyspike & slow waves, generalised slowing, and burst-suppression [16]. The effect of carbogen on NCSE EEG is not well understood, however. To quantify changes in EEG dynamics, several methods are available. Band-power analysis shows EEG signal power in particular frequency bands, and functional connectivity–the inference of brain networks by measuring signal similarity—has been used widely in epilepsy to show cortical network organisation before, during, and after seizures [16–21]. Graph theory metrics can further be applied to functional networks to assist interpretation, and elucidate specific aspects of the network. For example, clustering coefficient and path length are two graph theory measures commonly used to quantify local and global properties of a network [22]. These two measures have also been shown to vary over the course of a seizure [23].

In this exploratory study we investigated the effect of carbogen on band power and functional connectivity across five frequency sub-bands (delta, theta, alpha, beta and gamma). We hypothesised medium to large (cohen's *d >0.5*) quantitative EEG changes during- and after-carbogen administration.

## 2. Methods

The study design and the data are available from Forsyth et al [7] as this is the follow up study. The methods are organized in five main sections: (1) patient information and the corresponding EEG recordings, (2) data pre-processing, (3) band-power analysis, (4) graph-theoretical measures, and (5) statistical analysis. Fig 1 summarises the whole analysis pipeline.

### 2.1 Patient information and recordings

The study received full Research Ethics approval from the UK NHS Health Research Authority research ethics service (North East Tyne & Wear South Research Ethics Committee; reference 12/NE/0005, 1804/2020) and was registered as clinical trial (EudraCT 2011-005318-12). The approach to obtaining consent from families aimed to avoid time pressure in reaching a decision. If NCSE was a diagnostic possibility and an EEG was planned, the study was explained to families and an unhurried opportunity for questions was offered, before parents provided

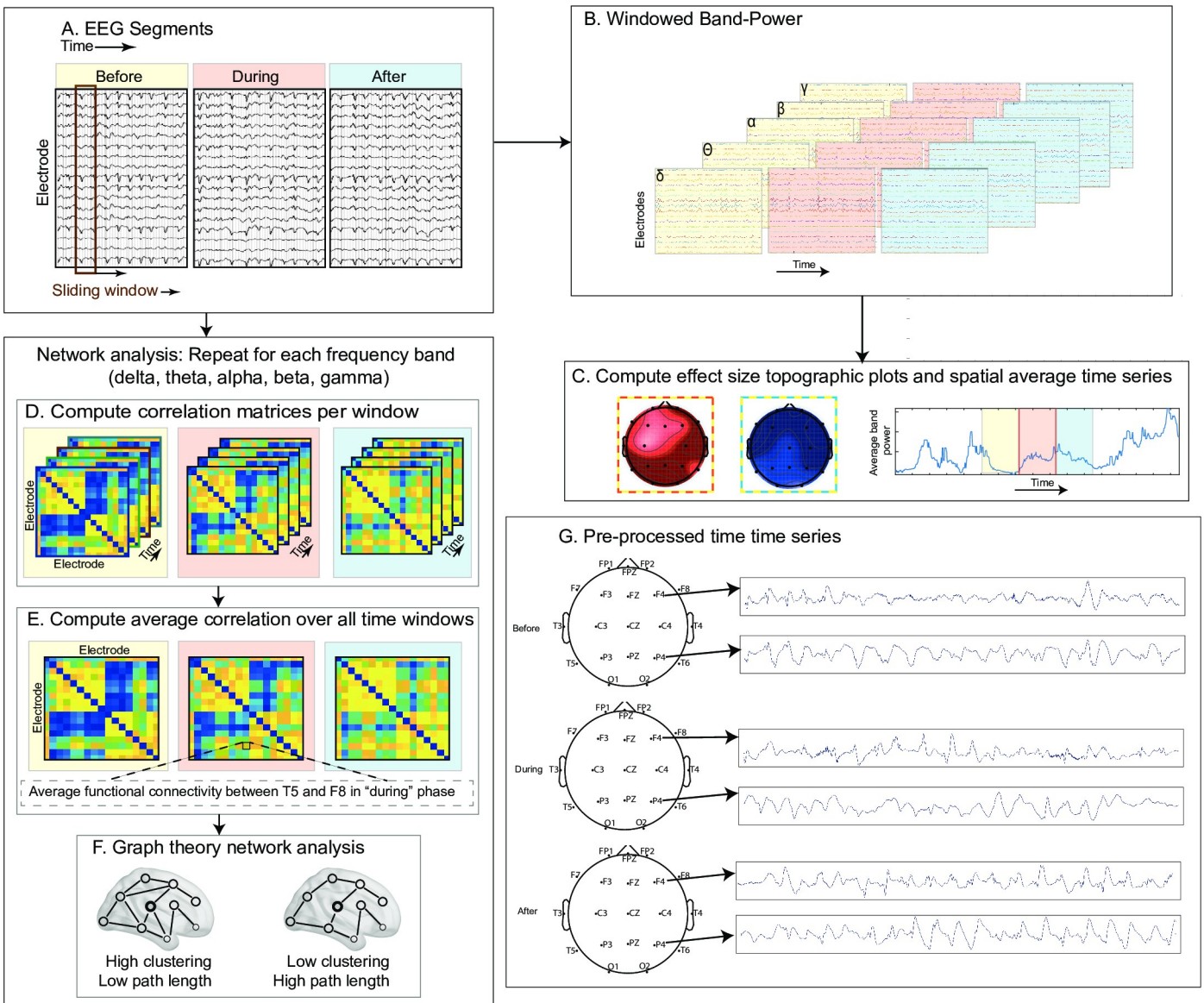

**Fig 1. Analysis Pipeline (A-F).** (A). Pre-processed EEG signal in "before", "during" and "after" states (B). Band-power computed across all the channels for every sliding window (C). Windowed band-power across all the channels used for topographic and spatial average time series plots. (D-E) windowed correlations matrices for every sliding window and then averaged over time (F). Illustrative networks demonstrating path length and clustering coefficient. (G) Pre-processed EEG time series for two example locations for Patient 3.

written consent for participation in principle in the study, in the understanding that at that point NCSE was not confirmed. The child then was transferred to the EEG department and neurophysiological monitoring commenced. Only if the EEG confirmed NCSE however did the child receive carbogen [7].

Patients selected for this study had all been diagnosed with epilepsy with either known aetiologies or different underlying neurodevelopmental disabilities. Inclusion criteria included (i) confirmed NCSE, with EEG manifestation, and (ii) reduced awareness or function confirmed by a parent or carer. Patients requiring other urgent treatment, or patients with capillary pCO2>8kPa were excluded. Patient recruitment in the prospective trail was very slow

**Table 1. Clinical and demographic information of all patients.**

| Patient | Gender | Age [yrs] | Aetiology | Number of Electrodes | Sampling rate [Hz] | Pre-inhalation EEG-Duration [s] | Post-inhalation EEG-Duration [s] |
|---|---|---|---|---|---|---|---|
| 1 | M | 4 | Angelman-Syndrom | 20 | 500 | 1200 | 1158 |
| 2 | F | 10 | Lissencephaly | 19 | 256 | 1200 | 979 |
| 3 | M | 3 | Lissencephaly (PAFAH1B1 mutation) | 20 | 256 | 416 | 1128 |
| 4 | M | 13 | Alper mitochondrial depletion syndrom (POLG1 mutation) | 20 | 256 | 989 | 1532 |
| 5 | M | 5 | Angelman | 20 | 256 | 1200 | 1224 |

despite opening the recruitment from additional centres. The recruitment was closed by Trial steering committee 30 months after recruiting the first child. This is done on the basis that substantial increases in recruitment rates were unrealistic. Forsyth et al [7] recruited six subjects, however, quality EEG recordings are available only for five of them. The mean age of the sample is 7± 3.85 years (mean ± standard deviation), of which four patients were male. Children in NCSE were given high flow inhaled carbogen by face mask for maximum 120s (113s±7s) with concurrent EEG measurement. All recordings were performed with a standard 10–20 clinical recording system [7]. EEG recording commenced a minimum of 10 minutes before commencing carbogen inhalation which was for 120s [7]. The clinical and demographic information of all patients in this study is summarised in Table 1.

Patient EEG activity was recorded continuously 1001.08 ± 303.55s pre-inhalation, during inhalation, and 1204.3 ± 182.45s post carbogen inhalation. A provision in the protocol allowed for the premature discontinuation of inhalation if the clinicians (including parents) felt it was any distress [7]. The data were collected under to ethical guidelines and protocols were monitored by a responsible clinician.

## 2.2 Data pre-processing

The data was notched at 50Hz (to exclude power line interference), and band-pass filtered between 1 to 70Hz using forward and backward $2^{nd}$ order Butterworth filter. Data was visually inspected for amplifier saturations or noisy channels; the eye blink-artefacts were rejected using Independent Component Analysis (ICA). For data analysis we extracted three epochs from the EEG data of every patient: "before" (immediately before inhalation of carbogen), "during" (during inhalation of carbogen) and "after" (immediately after inhalation of carbogen). The length of each epoch in "before" and "after" state is 120s. The length of epoch in "during" state varies between 106s-120 across the patients.

## 2.3 Band-power analysis

A two second sliding window with 50% overlap was used to extract absolute spectral power in five different frequency bands (delta: 1-4Hz, theta: 4–8 Hz, alpha: 8–13 Hz, beta: 13–30 Hz, and gamma: 30-70Hz) across each channel. The mean absolute powers (and the standard deviations) in each sliding window were averaged across windows for each frequency band, channel and patient. This gives five features per channel across three different states (before, during, and after) for each patient. This is visually summarised in Fig 1A–1C. The spatial average time series (average over all the channels) in broadband (1-70Hz) has also been plotted in Fig 2 across three states to visualise the effect of carbogen over time. The data was smoothed using a moving median filter (n = 31), to minimize the fluctuation of the signal for visualisation purpose, however, all the statistical calculations were carried out on original non-smoothed signal (S1 Fig).

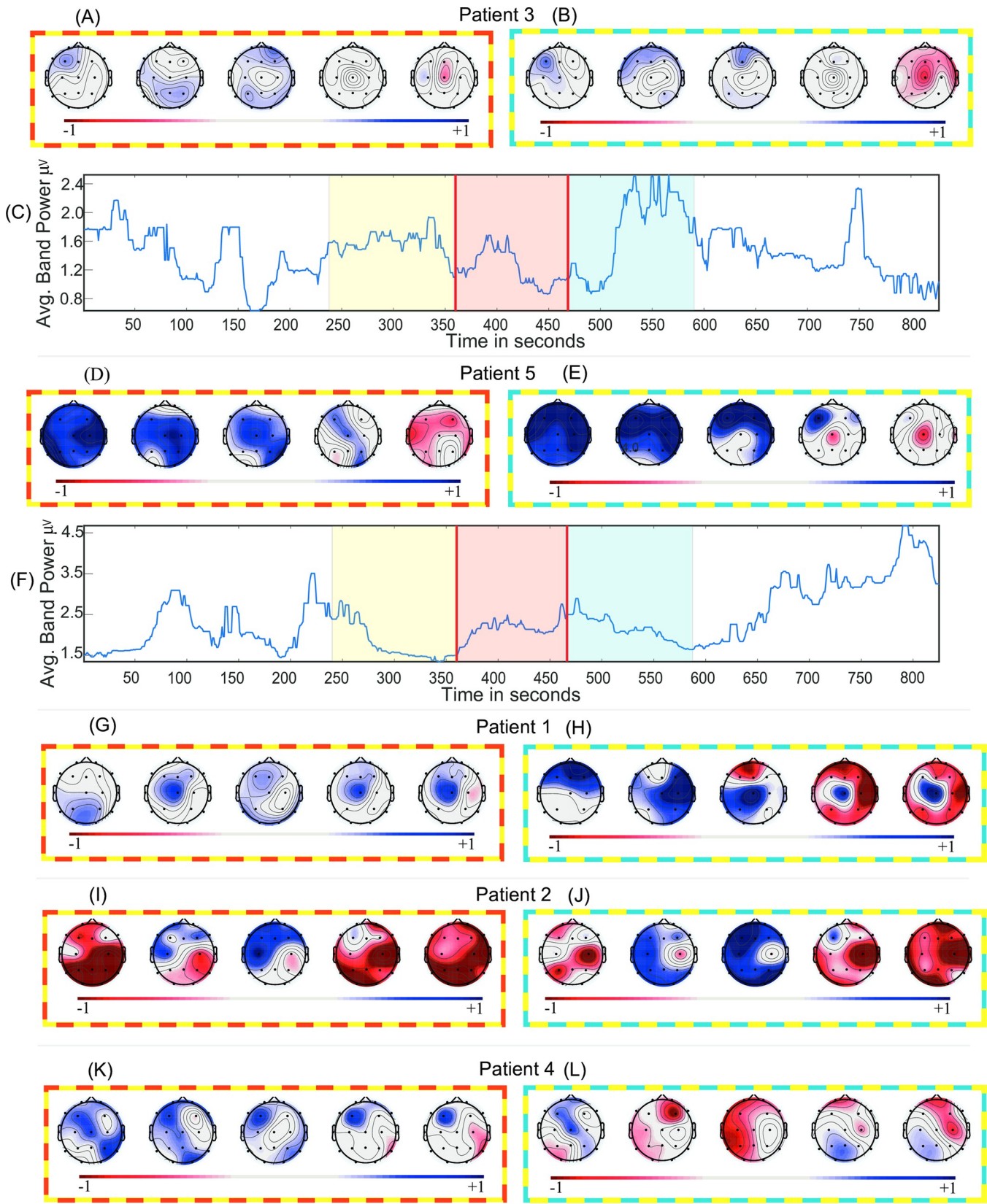

**Fig 2. Band-power analysis.** (A, D, G, I, K) Effect sizes (Cohen's *d*) of the band power change of "before-during" and (B, E, H, J, L) "before-after" across five frequency bands. (C, F) Time series of the average band-power in broadband (1-70Hz) for Patient 3 and Patient 5 respectively. Positive effect size denotes decrement in absolute band power. Topographical maps are ordered as delta, theta, alpha, beta and gamma. The frames surrounding topographical maps signify difference in the respective states they are representing. For instance, Fig 2C has yellow, red, and blue colours over laid across before, during, and after epochs respectively and topographical maps in Fig 2A has "yellow-red" colour frame representing the effect size between "before-during" states.

## 2.4 Network construction & graph theory analysis

The same EEG segments (before, during, after) used in band-power analysis were also used here to calculate the functional connectivity (amplitude correlation) and graph theory measures. The data has been bandpass filtered in the above-mentioned frequency bands followed by a sliding window analysis, as in band-power analysis. Each window resulted in a functional connectivity matrix (absolute Pearson's correlation matrix and diagonal correlation values were set to zero) through which graph theory measures (clustering coefficient and path length) were calculated using Brain Connectivity Toolbox [24]. This is visually depicted in Fig 1D–1F. The spatial average broadband functional connectivity, path length and clustering coefficient time series have been plotted in Fig 3 across three states to visualise the effect of carbogen across time.

The data was smoothed using a moving median filter (n = 31), to minimize the fluctuation of the signal for visualisation purpose, however, all the statistical calculations were carried out on original non-smoothed signal which can be found in S2–S4 Figs.

## 2.5 Statistical analysis

The effect size (Cohen's *d*) comparing the states "before" to "during", and "before" to "after" across all the channels and frequency bands was plotted in topographical plots in Fig 2. Additionally, we calculated the actual percentage change (Eq (1)) of the band-power from "before" to "during" and "before" to "after" states across all the electrodes for each patient in each frequency band. The percent changes are summarised in S5 Fig.

$$C = \frac{(\text{BPpost} - \text{BPpre})}{(\text{BPpost} + \text{BPpre})} \tag{1}$$

BPpost is band−power in "during" or "after" states.

BPpre is band−power in "before" state.

The effect sizes were also calculated for the functional connectivity in each entry of the matrix. Only medium and large effect sizes (threshold > = 0.5) are displayed in Fig 3. The time varying net broadband functional connectivity (mean of connectivity matrix across every sliding window) is plotted in Fig 3.

Permutation test (10,000 permutations; mean) and Cohen's *d* were used to quantify the effect of carbogen on patient's EEG measures (band-power and functional connectivity). MATLAB was used to perform the above mentioned statistical tests.

## 3. Results

The results section is divided into two sections. First, the band-power analysis and second, the functional network analysis. For brevity, two example patients' results are presented in full, with the remaining in the supplementary material.

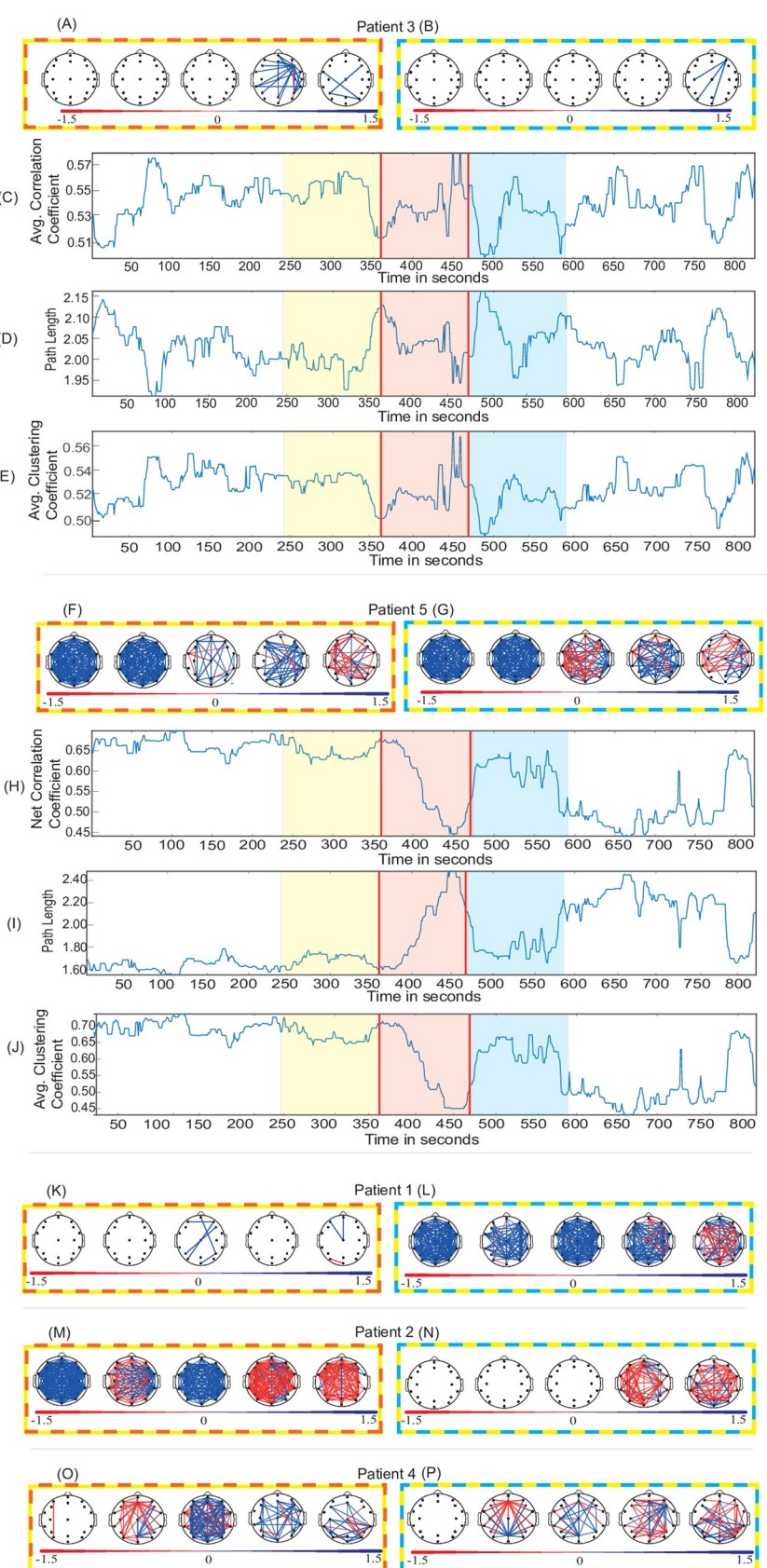

**Fig 3. Functional connectivity and graph-theory analysis.** (A, F, K, M, O) Effect sizes (Cohen's *d*) of the functional connectivity change of "before-during" and (B, G, L, N, P) "before-after" for five Patients. (C, H) Time series of net (average) correlation coefficient in broadband (1-70Hz) for Patient 3 and Patient 5. (D, I) Time series of the net (average) path length in broadband (1-70Hz) for Patient 3 and Patient 5. (E, J) Time series of the net (average) clustering coefficient in broadband (1-70Hz) for Patient 3 and Patient 5. Positive effect size denotes decrement in functional connectivity. Topographical maps are ordered as delta, theta, alpha, beta and gamma. The frames surrounding topographical maps signify difference in the respective states they are representing. For instance, Fig 3C has yellow, red, and blue colours over laid across before, during, and after epochs respectively and topographical maps in Fig 3A has "yellow-red" colour frame representing the effect size between "before-during" states.

## 3.1 Band power analysis

The effect size of the band-power across all channels in each frequency sub-band for each patient in the "before-during" and "before-after" comparison is summarised in Fig 2A, 2B, 2D, 2E, 2G–2L. The spatial average broadband time series and effect sizes show no clear or consistent pattern across all patients. In the "before-during" comparison, a positive effect indicates decrement of absolute band-power in "during" state relative to "before" state, whilst an increment indicates the opposite. The above statement also applies to the "before-after" comparison.

Fig 2C shows the net (spatial average) broadband time series for Patient 3. The effect sizes of "before-during" and "before-after" comparisons indicate small ($d = 0.32$, $p = 0.03$; $d = -0.17$, $p = 0.30$) effects. The effect sizes in "before-during" comparisons (Fig 2A) indicate a small effect across all the frequency bands. The same is observed in the "before-after" comparison with an exception in the gamma band. The effect size and *p*-values across every channel for "before-during" and "before-after" comparisons are summarised in S5 and S6 Tables. This result indicates a small effect of carbogen on the EEG band-power of Patient 3.

In contrast to Patient 3, large effects are observed in "before-during" and "before-after" comparisons for the EEG in Patient 5 (summarised in plots in Fig 2D and 2E). These effect sizes indicate the suppression of band-power especially in the lower frequency bands during and after carbogen inhalation. The effect size and *p*-values are summarised in S9 and S10 Tables. The suppression of band-power is visible in the sub-bands (Fig 2D and 2E), however, it cannot be observed in the average broadband time series in Fig 2F ("before-during": $d = -0.14$; $p = 0.47$ and "before-after": $d = 0.01$; $p = 0.96$).

The two representative patients: Patient 3 and Patient 5 indicates contrasting effects of carbogen across all the sub-bands and broadband. This non-homogenous effect of the carbogen on absolute band-powers is also observable across three other patients (Fig 2G–2L). The net broadband time series (for the remaining patients; S2F–S2H Fig) and normalised sub-band time series (for all the patients; A-E) can be found in S6 Fig. S11 and S12 Tables and S1–S10 Tables contain the effect sizes and permutation test *p*-values for broadband time series and individual channels across all the sub-bands respectively.

## 3.2 Network analysis

**3.2.1 Functional connectivity.** In Patient 3 (Fig 3A and 3B), the functional connectivity in the "before-during" comparison does not show any medium or large effects ($d > = 0.5$) of the carbogen in the delta, theta or alpha sub-bands. Beta and gamma bands show reduction of the connectivity between few nodes. In the "before-after" (Fig 3B) comparison, carbogen shows only a small effect in the gamma band. The insignificance ($d = 0.03$, $p = 0.82$) of carbogen inhalation in "during" state, and significant effect on post inhalation ($d = 0.31$, $p = 0.03$) can also be seen in Fig 3C across the broadband time series.

Patient 5 on the other hand, shows a clear suppression in the net broadband functional connectivity when comparing "during" to "after" states (Fig 3H). This indicates a significant effect

during carbogen inhalation ($d = 0.82$, $p<0.001$) and a small but significant effect in the "after" inhalation state ($d = 0.42$; $p<0.001$). The topographical plots in patient 5 (Fig 3F and 3G) also show contrasting results in comparison to Patient 3. In Fig 3F, the functional connectivity is suppressed in lower frequencies during carbogen inhalation, however, alpha and beta bands are less affected and also show some increment in connectivity. In the higher frequencies (gamma), the predominant effect of elevation in the connectivity between most nodes can be observed in the "before-during" comparison. The same observations can be made in the "before-after" comparison (Fig 3G).

The results of functional connectivity analysis of the other three patients also suggests this non-homogenous influence of carbogen on the EEG. The sub-band functional connectivity (S7A–S7E Fig) and broadband time series (S7F–S7H Fig) results for the remaining subjects are summarised in supplementary materials. The effect size and permutation test *p*-values for the remaining subjects can be found in S11 and S12 Tables respectively in supplementary materials.

**3.2.2 Graph theory.**   The path length and average clustering coefficient in broadband for two representative patients is plotted as a time series in Fig 3D, 3E, 3I, 3J. In Patient 3, a significant ($d = -0.36$, $p = 0.01$) effect is observed in the "after" states (Fig 3D) in path length only. The net clustering coefficient also exhibits the similar property as path length.

In contrast to Patient 3, in Patient 5 (Fig 3I), the net path length has increased significantly in "during" ($d = -0.88$, $p = 0.00$) and "after" states ($d = -0.41$, $p = 0.004$; Fig 3I). However, clustering coefficient has been significantly supressed "during" ($d = 0.81$, $p = 0.00$) and "after" ($d = 0.39$, $p = 0.007$) inhalation of carbogen (Fig 3J). The results for the remaining subjects also show inconsistency in the effect of carbogen amongst them. The sub-band (S8A–S8E Fig) and broadband path length time series (S8F–S8H Fig) for the remaining subjects are summarised in supplementary materials. The sub-band (S9A–S9E Fig) and broadband clustering coefficient time series (S9F–S9H Fig) results for the remaining subjects are summarised in supplementary materials.

# 4. Discussion

The present work investigates the relationship between carbogen inhalation and quantitative EEG measures. Specifically, we compared spectral power, functional connectivity and graph theory measures "before" to "during" and "after" the inhalation of carbogen in paediatric NCSE. For these methods we did not find evidence to suggest a common effect of carbogen on NCSE EEG.

Lennox [9] applied 10% $CO_2$ to successfully supress the spike wave EEG activity. This was extended in several studies on rodent, canine and non-human primate models [10–12, 15, 25, 26]. 15–30% $CO_2$ prevented electrically induced convulsions in psychiatric patients [10]. Tolner et al [8] in their pilot study on seven patients with drug resistant epilepsy used standard medical carbogen (5% $CO_2$) and reported the rapid termination of electrographic seizures despite the fact that the application of carbogen was started after the seizure generalization. In the above study, none of the patients had any underlying neural disability except "epilepsy". This is important to note because the cohort in our study has mixed aetiology.

The inhalation of carbogen induces a mild temporary acidosis (i.e. lowering pH) and when reduced to a critical value (blood pH-level<7.35) this is termed as acidaemia [27]. Acidosis attenuates excitatory neurotransmission by reducing NMDA-receptor activity [28] whilst enhancing inhibitory neurotransmission by facilitating the responsiveness of GABAA receptors [28]. Possible cellular mechanisms [29] involve direct effects of pH on voltage and ligand-gated ion channel conductance [15, 30, 31] and adenosine signalling [32]. Therefore, one may expect a decrease in the EEG band-power, which was also reported previously [33].

In Patient 3, no substantial effects were observed in "during" and "after" states relative to the "before" state. However, in Patient 5, carbogen delivery was associated with reduced correlation between the nodes (i.e. positive effect size) in lower frequency bands. This may be due to the fact that acidosis introduced by the carbogen might not be sufficient enough to supress the high frequency activity. A small, and insignificant change of path length and clustering coefficient in during state, can be observed in Patient 3. This might be due to the fact that carbogen was inhaled not at the seizure onset but very late into the seizure causing a delayed acidosis effect. This delayed effect might not be enough to terminate the seizure state and show change in EEG recordings. Application of carbogen immediately after the seizure onset might have strong anticonvulsant and therapeutic interventions [8].

Overall the effect of the carbogen on patient EEG was inconsistent across all measurements used. Note that although in our analysis no measure showed a consistent effect, this does not prohibit the possibility of the existence of a measure showing consistent effects across patients. In other words, absence of evidence does not necessarily represent evidence of absence and our finding should therefore be considered with this in mind. Our choices of measures are routine in the field of quantitative EEG analysis and have been shown to demonstrate differences in a wide range of settings [22, 34–37]. Other factors for the ineffectiveness of carbogen might be, first, the lack of "aggressiveness" (concentration/duration), second, the heterogenety of the underlying aetiology of the patients, and third our limited sample size without a control arm of non-delivery of carbogen.

Even though carbogen has the potential to terminate NCSE seizures in humans, our results suggest possibly different effects in different subjects. This heterogeneity may be related to underlying aetiology, which should be considered in future studies.

## Supporting information

**S1 Fig.** (A)-(E) Non-smoothed band power time series in broadband across three different states for five patients.
(DOCX)

**S2 Fig. Non-smoothed correlation coefficient in broadband across three different states for five patients.**
(DOCX)

**S3 Fig. Non-smoothed path length time series in broadband across three different states for five patients.**
(DOCX)

**S4 Fig. Non-smoothed clustering coefficient time series in broadband across three different states for five patients.**
(DOCX)

**S5 Fig. Percentage change in band power.** The normalised percentage change in band power across different frequencies between A) "Before-During" state and B) "Before-After" state.
(DOCX)

**S6 Fig. Band power time series.** (A)-(E) Sub-band normalised band power time series for Patient 1-Patient 5. (F)-(H) Broandband average time series for Patient 1, Patient 2 and Patient 4.
(DOCX)

**S7 Fig. Correlation coefficient time series.** (A)-(E) Sub-band normalised correlation coefficient time series for Patient 1-Patient 5. (F)-(H) Broandband average time series for Patient 1, Patient 2 and Patient 4.
(DOCX)

**S8 Fig. Path length time series.** (A)-(E) Sub-band path length time series for Patient 1-Patient 5. (F)-(H) Broandband path length time series for Patient 1, Patient 2 and Patient 4.
(DOCX)

**S9 Fig. Clustering coefficient time series.** (A)-(E) Sub-band normalised clustering coefficient time series for Patient 1-Patient 5. (F)-(H) Broandband average time series for Patient 1, Patient 2 and Patient 4.
(DOCX)

**S1 Table. Patient 1 Effect size (Effect size *d*-values) for all the channels across all the frequency bands for before-during and before-after state.**
(DOCX)

**S2 Table. Patient 1 Permutation test p-values (FDR corrected) for all the channels across all frequency sub-bands in before-during and before-after state.**
(DOCX)

**S3 Table. Patient 2 Effect size (Effect size *d*-values) for all the channels across all the frequency bands for before-during and before-after state.**
(DOCX)

**S4 Table. Patient 2 Permutation test p-values (FDR corrected) for all the channels across all frequency sub-bands in before-during and before-after state.**
(DOCX)

**S5 Table. Patient 3 Effect size (Effect size *d*-values) for all the channels across all the frequency bands for before-during and before-after state.**
(DOCX)

**S6 Table. Patient 3 Permutation test p-values (FDR corrected) for all the channels across all frequency sub-bands in before-during and before-after state.**
(DOCX)

**S7 Table. Patient 4 Effect size (Cohen's *d*–values) for all the channels across all the frequency bands for before-during and before-after state.**
(DOCX)

**S8 Table. Patient 4 Permutation test p-values (FDR corrected) for all the channels across all frequency sub-bands in before-during and before-after state.**
(DOCX)

**S9 Table. Patient 5 Effect size (Cohen's *d*–values) for all the channels across all the frequency bands for before-during and before-after state.**
(DOCX)

**S10 Table. Patient 5 Permutation test p-values (FDR corrected) for all the channels across all frequency sub-bands in before-during and before-after state.**
(DOCX)

**S11 Table. Effect size (Cohen's *d*–values) for broadband power time series, functional connectivity time series, path length time series, and clustering coefficient time series.** (DOCX)

**S12 Table. Permutation test p-values for broadband band power time series, functional connectivity time series, path length time series, and clustering coefficient time series.** (DOCX)

## Acknowledgments

None of the authors have any conflict of interest. The funders had no role in the study design, data collection and analysis, decision to publish or preparation of the manuscript.

## Author Contributions

**Conceptualization:** S. Ramaraju.

**Data curation:** S. Ramaraju.

**Formal analysis:** S. Ramaraju.

**Funding acquisition:** R. Forsyth, P. N. Taylor.

**Methodology:** S. Ramaraju.

**Resources:** R. Forsyth, P. N. Taylor.

**Supervision:** Y. Wang, R. Forsyth, P. N. Taylor.

**Visualization:** S. Ramaraju, S. Reichert.

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
