## [Decision Letter · Decision Letter 0]

3 Mar 2020

PONE-D-19-31394

Carbogen inhalation during Non-Convulsive Status Epilepticus: A quantitative analysis of EEG recordings

PLOS ONE

Dear Dr Ramaraju,

Thank you for submitting your manuscript to PLOS ONE. After careful consideration, we feel that it has merit but does not fully meet PLOS ONE’s publication criteria as it currently stands. Therefore, we invite you to submit a revised version of the manuscript that addresses the points raised during the review process.

ACADEMIC EDITOR:Although the topic is quite novel I share the main criticisms by both reviewers that. The authors should perform the quantitative characterization in relation to outcome. In addition Ethics approval for the retrospective study should be obtained;Inclusion and exclusion criteria should be specified; which EEG procedure was used to test Carbogen efficacy? Statistical analysis: should be explained; 

We would appreciate receiving your revised manuscript by Apr 17 2020 11:59PM. To enhance the reproducibility of your results, we recommend that if applicable you deposit your laboratory protocols in protocols.io, where a protocol can be assigned its own identifier (DOI) such that it can be cited independently in the future. For instructions see: http://journals.plos.org/plosone/s/submission-guidelines#loc-laboratory-protocols

We look forward to receiving your revised manuscript.

Kind regards,

Andrea Romigi, M.D., Ph.D

Academic Editor

PLOS ONE

Journal Requirements:

2. Please state in your methods section whether you obtained consent from parents or guardians of the minors included in the study or whether the research ethics committee or IRB approved the lack of parent or guardian consent.

3. Thank you for including your ethics statement:

'The study received full ethics review and approval (REC reference 12/NE/0005) and was registered as clinical trail (EudraCT 2011-005318-12). Because final confirmation of eligibility required confirmation of NCSE on EEG, written “consent in principle” to participate in the study was obtained with final verbal consent to proceed once NCSE was confirmed electrographically. '

a. Please amend your current ethics statement to include the ** full name ** of the ethics committee/institutional review board(s) that approved your specific study.

Reviewers' comments:

Reviewer's Responses to Questions

**Comments to the Author**

1. Is the manuscript technically sound, and do the data support the conclusions?

Reviewer #1: Partly

Reviewer #2: Partly

2. Has the statistical analysis been performed appropriately and rigorously? 

Reviewer #1: No

Reviewer #2: No

3. Have the authors made all data underlying the findings in their manuscript fully available?

Reviewer #1: Yes

Reviewer #2: Yes

4. Is the manuscript presented in an intelligible fashion and written in standard English?

Reviewer #1: Yes

Reviewer #2: Yes

5. Review Comments to the Author

Reviewer #1: This is a report of the study on carbogen inhalation during Non-Convulsive Status Epilepticus: A quantitative analysis of EEG recordings.

Below are some comments:

- The objective of the study was to ‘quantify the effect of inhaled 5% carbon-dioxide/95% oxygen on EEG recordings from patients in non-convulsive status epilepticus (NCSE)’. The objective was however, not followed by a precise/clear hypothesis. Neither did the authors mention how the sample size of 5 patients derived (no description on how the study was powered, and no description on what was the targeted effect size).

- In addition to small sample size of 5 patients, patients enrolled in the study had quite diverse etiology and hence introducing heterogeneity in the selected population. This raises some questions on the usefulness of the current analysis

- Patients selection was retrospective and this added heterogeneity in the selected samples/patients. Normally in such a retrospective study, larger sample size is expected to compensate for the heterogeneity due to not being able to make a proper selection of patients. Again this restriction in addition to very small sample size should be explained and addressed.

- Restriction of the retrospective analysis also applies in the treatment onset, as there were indications that some patients may receive the treatment (carbogen) in different time frame (onset) than other. This adds further heterogeneity in the collected data.

- Lacking of control arm (i.e. patients not receiving any treatment) in this analysis make the data interpretation is difficult. Even though the nature of the data is longitudinal/time series in which some parts in each patient exposed to treatment and the rest not, which implies that patient can become their own control for different phases they involved during the study, but in the event of inconsistent results as currently observed, the existence of non-exposed patients might have helped the interpretation.

- The time series analysis may have quite large data on EEG collected longitudinally, yet the experimental unit of this study is patient. With this restriction, it is difficult if possible at all to make any conclusion from this analysis.

- Page 8: ‘Because final confirmation of eligibility required confirmation of NCSE on EEG, written “consent in principle” to participate in the study was obtained with final verbal consent to proceed once NCSE was confirmed electrographically’. How many patients were screened out and declared ineligible?

Reviewer #2: Manuscript number: PONE-D-19-31394

Reviewer’s Comments:

The manuscript presents a quantitative analysis of EEG recordings of carbogen inhalation during NCSE. The study described in this manuscript is interesting but the following points need to be addressed:

1) quantitative characterization should be performed in relation to outcome

2) Ethics approval for the retrospective study should be obtained

3) inclusion and exclusion criteria are missed

4) EEG procedures should be added

5) Statistical analysis: please specify which tests were applied for comparisons and the statistical software used

6. PLOS authors have the option to publish the peer review history of their article (what does this mean?). If published, this will include your full peer review and any attached files.

Reviewer #1: No

Reviewer #2: No

---

## [Author Response · Author response to Decision Letter 0]

24 May 2020

Reviewer #1: 

This is a report of the study on carbogen inhalation during Non-Convulsive Status Epilepticus: A quantitative analysis of EEG recordings.

Below are some comments:

We thank the reviewer for their review and comments which we address below.

- The objective of the study was to ‘quantify the effect of inhaled 5% carbon-dioxide/95% oxygen on EEG recordings from patients in non-convulsive status epilepticus (NCSE)’. The objective was however, not followed by a precise/clear hypothesis. 

The hypothesis is now added to the manuscript in the introduction. “We hypothesised quantitative EEG changes during- and after-carbogen administration.”

Neither did the authors mention how the sample size of 5 patients derived (no description on how the study was powered, and no description on what was the targeted effect size).

We used all data from all patients with pre- during- and post-carbogen administration available from Forsyth et al (2016), which contains further details pertaining to study power. We wish to highlight that the 2016 study was originally to evaluate the safety and efficacy of low dose carbogen administration. Our present study is an opportunistic retrospective analysis of that unique and rare prospective clinical trial data. 

- In addition to small sample size of 5 patients, patients enrolled in the study had quite diverse etiology and hence introducing heterogeneity in the selected population. This raises some questions on the usefulness of the current analysis

We acknowledge the reviewer’s comment. We include the following in discussion.

“Other factors for the ineffectiveness of carbogen might be, first, the lack of “aggressiveness” (concentration/duration), second, the heterogeneity of the underlying aetiology of the patients, and third our limited sample size.”

- Patients selection was retrospective and this added heterogeneity in the selected samples/patients. Normally in such a retrospective study, larger sample size is expected to compensate for the heterogeneity due to not being able to make a proper selection of patients. Again this restriction in addition to very small sample size should be explained and addressed.

The following is now added to the manuscript.

“Patient recruitment in this prospective trial was slow and despite recruiting from other centres a sample of only five patients was possible.”

- Restriction of the retrospective analysis also applies in the treatment onset, as there were indications that some patients may receive the treatment (carbogen) in different time frame (onset) than other. This adds further heterogeneity in the collected data.

Agreed, we acknowledge this as a one of the limitations of this study. This was a prospective trial with a novel drug which hasn’t been used in this circumstance so we presented results in a cautious manner.

- Lacking of control arm (i.e. patients not receiving any treatment) in this analysis make the data interpretation is difficult. Even though the nature of the data is longitudinal/time series in which some parts in each patient exposed to treatment and the rest not, which implies that patient can become their own control for different phases they involved during the study, but in the event of inconsistent results as currently observed, the existence of non-exposed patients might have helped the interpretation.

We appreciate the concerns in the interpreting the data and therefore we present the results as in an exploratory fashion. We used each patient’s non-exposed time frame as control signal. We acknowledge the reviewer’s suggestion of using non-exposed patients as controls, however, the outcome of current analysis states that except one patient, remaining cohort does not show any effect on EEG. 

- The time series analysis may have quite large data on EEG collected longitudinally, yet the experimental unit of this study is patient. With this restriction, it is difficult if possible at all to make any conclusion from this analysis.

We acknowledge with the reviewer’s comment. We did perform the analysis on individual EEG channels across three phases (before, during and after phases of carbogen inhalation) and can be found in supplementary materials. The results are not drastically different from the patient level results in terms of inconsistency.

We do think it is reasonable to conclude that we found no evidence to support the hypothesis that 5% carbogen administration is associated with large and consistent alterations to commonly used quantitative EEG metrics in this patient cohort.

- Page 8: ‘Because final confirmation of eligibility required confirmation of NCSE on EEG, written “consent in principle” to participate in the study was obtained with final verbal consent to proceed once NCSE was confirmed electrographically’. How many patients were screened out and declared ineligible?

There was one child in whom NCSE was suspected (on basis of known seizure disorder and parental report of reduced alertness and interaction) but who in fact turned out not to be in NCSE once the EEG was in process and was therefore subsequently excluded.

 

Reviewer #2: 

Manuscript number: PONE-D-19-31394

Reviewer’s Comments:

The manuscript presents a quantitative analysis of EEG recordings of carbogen inhalation during NCSE. The study described in this manuscript is interesting but the following points need to be addressed:

We thank the reviewer for their review and comments which we address below.

1) quantitative characterization should be performed in relation to outcome

In terms of clinical outcome all patients did not respond favourably (i.e. seizure persistence despite carbogen administration). In terms of quantitative EEG changes, which was our variable of interest has shown large effect on patient 5’s EEG (suppression of bandpower in lower frequency bands) but has no effect of rest of the cohort’s EEG. 

2) Ethics approval for the retrospective study should be obtained

We have now included the ethical approval reference in the text (Ref: 1804/2020).

3) inclusion and exclusion criteria are missed

The following is now added to the manuscript.

“Inclusion criteria included (i) confirmed NCSE, with EEG manifestation, and (ii) reduced awareness or function confirmed by a parent or carer. Patients requiring other urgent treatment, or patients with capillary pCO2>8kPa were excluded.”

4) EEG procedures should be added

We have included EEG acquisition information including sampling rate and electrode numbers in Table 1.

5) Statistical analysis: please specify which tests were applied for comparisons and the statistical software used

Two statistical tests were applied for the comparisons: Permutation test (10000 permutations; mean) and Cohen’s d. Detailed information can be found in Statistical Analysis section.

MATLAB was used for the above-mentioned statistical tests. This is now added to the Statistical Analysis section.

---

## [Decision Letter · Decision Letter 1]

30 Jul 2020

PONE-D-19-31394R1

Carbogen inhalation during Non-Convulsive Status Epilepticus: A quantitative analysis of EEG recordings

PLOS ONE

Dear Dr. Ramaraju,

Thank you for submitting your manuscript to PLOS ONE. After careful consideration, we feel that it has merit but does not fully meet PLOS ONE’s publication criteria as it currently stands. Therefore, we invite you to submit a revised version of the manuscript that addresses the points raised during the review process.

Please provide replies for these further comments, in particular provide the sample size calculation and discuss the lack of the control arm.

We look forward to receiving your revised manuscript.

Kind regards,

Andrea Romigi, M.D., Ph.D

Academic Editor

PLOS ONE

Reviewers' comments:

Reviewer's Responses to Questions

**Comments to the Author**

1. If the authors have adequately addressed your comments raised in a previous round of review and you feel that this manuscript is now acceptable for publication, you may indicate that here to bypass the “Comments to the Author” section, enter your conflict of interest statement in the “Confidential to Editor” section, and submit your "Accept" recommendation.

Reviewer #1: (No Response)

Reviewer #2: (No Response)

2. Is the manuscript technically sound, and do the data support the conclusions?

Reviewer #1: Partly

Reviewer #2: Partly

3. Has the statistical analysis been performed appropriately and rigorously? 

Reviewer #1: Yes

Reviewer #2: No

4. Have the authors made all data underlying the findings in their manuscript fully available?

Reviewer #1: Yes

Reviewer #2: Yes

5. Is the manuscript presented in an intelligible fashion and written in standard English?

Reviewer #1: Yes

Reviewer #2: Yes

6. Review Comments to the Author

Reviewer #1: Thank you for the reply. Below please find some further comments in a couple of points:

- Lacking of control arm (i.e. patients not receiving any treatment) in this analysis make the data interpretation is difficult. Even though the nature of the data is longitudinal/time series in which some parts in each patient exposed to treatment and the rest not, which implies that patient can become their own control for different phases they involved during the study, but in the event of inconsistent results as currently observed, the existence of non-exposed patients might have helped the interpretation.

A: We appreciate the concerns in the interpreting the data and therefore we present the results as in an exploratory fashion. We used each patient’s non-exposed time frame as control signal. We acknowledge the reviewer’s suggestion of using non-exposed patients as controls, however, the outcome of current analysis states that except one patient, remaining cohort does not show any effect on EEG.

Re: The statement that the study is exploratory in nature is an important one and should be clearly mentioned in the manuscript.

- The time series analysis may have quite large data on EEG collected longitudinally, yet the experimental unit of this study is patient. With this restriction, it is difficult if possible at all to make any conclusion from this analysis.

A: We acknowledge with the reviewer’s comment. We did perform the analysis on individual EEG channels across three phases (before, during and after phases of carbogen inhalation) and can be found in supplementary materials. The results are not drastically different from the patient level results in terms of inconsistency.

We do think it is reasonable to conclude that we found no evidence to support the hypothesis that 5% carbogen administration is associated with large and consistent alterations to commonly used quantitative EEG metrics in this patient cohort.

Re: Hypothesis did not specify how large the changes/alterations were sought. With this non-specific hypothesis, the conclusion was also blurry and not specific. Therefore, the above stated conclusion is not fully supported. In your above conclusion, how do you define ‘large and consistent alterations’? This is particularly a concern since the sample size of the study was only 5 patients.

Reviewer #2: Please clearly indicate in the methods section the study design and that data are available from the study of Forsyth and collaborators (2016) with a brief description

Please add a brief description of the sample size calculation

Please add and discuss in the discussion the lack of the control arm

7. PLOS authors have the option to publish the peer review history of their article (what does this mean?). If published, this will include your full peer review and any attached files.

Reviewer #1: No

Reviewer #2: No

---

## [Author Response · Author response to Decision Letter 1]

18 Sep 2020

Re: The statement that the study is exploratory in nature is an important one and should be clearly mentioned in the manuscript

Authors: We agree with the reviewer’s suggestion. We now added “explored” in the last paragraph of introduction section (In this exploratory study we investigated the effect of carbogen….)

We have also updated the title to:

Carbogen inhalation during Non-Convulsive Status Epilepticus: A quantitative exploratory analysis of EEG recordings

Re: Hypothesis did not specify how large the changes/alterations were sought. With this non-specific hypothesis, the conclusion was also blurry and not specific. Therefore, the above stated conclusion is not fully supported. In your above conclusion, how do you define ‘large and consistent alterations’? This is particularly a concern since the sample size of the study was only 5 patients.

Authors: We have now been more precise in our hypothesis in the last paragraph of the introduction.

“In this exploratory study we investigated the effect of carbogen on band power and functional connectivity across five frequency sub-bands (delta, theta, alpha, beta and gamma). We hypothesised medium to large (cohen’s d >0.5) quantitative EEG changes during- and after-carbogen administration.”

Reviewer #2: Please clearly indicate in the methods section the study design and that data are available from the study of Forsyth and collaborators (2016) with a brief description

Authors: As reviewer suggested we now added the following sentence at the beginning of the Methods section. “The study design and the data are available from Forsyth et al (2016) as this is the follow up study”. 

Please add a brief description of the sample size calculation

Authors: The following paragraph is added in Patient information and recordings of Methods section

“Patient recruitment in the prospective trail was very slow despite opening the recruitment from additional centres. The recruitment was closed by Trial steering committee 30 months after recruiting the first child. This is done on the basis that substantial increases in recruitment rates were unrealistic. Forsyth et al (2016) recruited six subjects, however, quality EEG recordings are available only for five of them”.

Please add and discuss in the discussion the lack of the control arm

We have now included this in the limitations paragraph of discussion:

“…without a control arm of non-delivery of carbogen.”

---

## [Editor Report · Decision Letter 2]

29 Sep 2020

Carbogen inhalation during Non-Convulsive Status Epilepticus: A quantitative exploratory analysis of EEG recordings

PONE-D-19-31394R2

Dear Dr. Ramaraju,

We’re pleased to inform you that your manuscript has been judged scientifically suitable for publication and will be formally accepted for publication once it meets all outstanding technical requirements.

Kind regards,

Andrea Romigi, M.D., Ph.D

Academic Editor

PLOS ONE
---

## [Editor Report · Acceptance letter]

1 Dec 2020

PONE-D-19-31394R2 

Carbogen inhalation during Non-Convulsive Status Epilepticus: A quantitative exploratory analysis of EEG recordings 

Dear Dr. Ramaraju:

I'm pleased to inform you that your manuscript has been deemed suitable for publication in PLOS ONE. Congratulations! Your manuscript is now with our production department. 

Kind regards, 

on behalf of

Dr. Andrea Romigi 

Academic Editor

PLOS ONE